# Micronutrient Deficiencies Following Minimally Invasive Esophagectomy for Cancer

**DOI:** 10.3390/nu12030778

**Published:** 2020-03-15

**Authors:** Henricus J.B. Janssen, Laura F.C. Fransen, Jeroen E.H. Ponten, Grard A.P. Nieuwenhuijzen, Misha D.P. Luyer

**Affiliations:** Department of Surgery, Catharina Hospital, Eindhoven, Michelangelolaan 2, 5623 EJ Eindhoven, The Netherlands

**Keywords:** micronutrient deficiencies, vitamin deficiencies, esophagectomy, esophageal cancer

## Abstract

Over the past decades, survival rates for patients with resectable esophageal cancer have improved significantly. Consequently, the sequelae of having a gastric conduit, such as development of micronutrient deficiencies, become increasingly apparent. This study investigated postoperative micronutrient trends in the follow-up of patients following a minimally invasive esophagectomy (MIE) for cancer. Patients were included if they had at least one postoperative evaluation of iron, ferritin, vitamins B1, B6, B12, D, folate or methylmalonic acid. Data were available in 83 of 95 patients. Of these, 78.3% (65/83) had at least one and 37.3% (31/83) had more than one micronutrient deficiency at a median of 6.1 months (interquartile range (IQR) 5.4–7.5) of follow-up. Similar to the results found in previous studies, most common deficiencies identified were: iron, vitamin B12 and vitamin D. In addition, folate deficiency and anemia were detected in a substantial amount of patients in this cohort. At 24.8 months (IQR 19.4–33.1) of follow-up, micronutrient deficiencies were still common, however, most deficiencies normalized following supplementation on indication. In conclusion, patients undergoing a MIE are at risk of developing micronutrient deficiencies as early as 6 up to 24 months after surgery and should therefore be routinely checked and supplemented when needed.

## 1. Introduction

Esophageal cancer has always been associated with a poor survival, but over the past decades the survival rate for patients undergoing surgery has significantly improved with the introduction of neoadjuvant chemoradiotherapy [1,2,3]. Improved perioperative care, the implementation of enhanced recovery programs and the introduction of minimally invasive techniques, have further contributed to a reduced postoperative mortality and morbidity [4,5,6,7,8]. With these improvements and an increase in survival rates, patients’ quality of life and long-term sequelae—such as development of micronutrient deficiencies—of this intensive treatment are becoming more relevant [9,10,11,12].

In bariatric [13,14,15,16,17] and gastric cancer surgery [18,19,20,21], it has been shown that new micronutrient deficiencies may develop on the long-term or pre-existent deficiencies worsen postoperatively in a considerable number of patients. This effect was most prominent following bariatric Roux-en-Y gastric bypass surgery and total gastrectomy for cancer with Roux-en-Y esophagojejunostomy. Clinically relevant micronutrient deficiencies commonly found in these studies include iron and vitamins A, B1, B12, D and E. It has been hypothesized that micronutrient deficiencies may also play an important role in quality of life after an esophagectomy for cancer, due to the anatomical changes made to the upper gastrointestinal tract. These anatomical changes, wherein the stomach is most commonly used as the conduit to replace the resected esophagus, often entail high morbidity [4,5,6,7]. Similar to bariatric surgery, these changes may be accompanied by adverse gastrointestinal symptoms, including nausea, vomiting, regurgitation, diarrhea, dumping or loss of appetite and reduced intake, which may contribute to further weight loss and malnutrition or anorexia [9,10,11,12]. 

Hence, the aim of this study is to review the literature of micronutrient deficiencies after an esophagectomy and to study micronutrient deficiencies in a specific cohort of patients after a minimally invasive esophagectomy.

## 2. Materials and Methods 

### 2.1. Search of the Literature

A literature search of the PubMed, MEDLINE and EMBASE databases was conducted by two authors (H.J. and J.P.) independently to identify the literature related to micronutrient deficiencies following an esophagectomy. Search terms included controlled terms from the MeSH database in PubMed and MEDLINE, the EMtree in EMBASE database as well as free text terms. Terms expressing “esophageal cancer”, “esophagectomy”, “vitamin deficiencies”, ‘‘micronutrient deficiencies”, “avitaminosis”, “hypovitaminosis”, “malabsorption”, “malnutrition” and “metabolic complications” in all databases, were variously combined in the search. Reference lists were screened and forward citation tracking was conducted to identify literature missed by the initial search. All full-text articles written in English that reported on vitamin deficiencies or vitamin status were included in the review.

### 2.2. Study Design and Cohort

The second part of this study investigated the postoperative micronutrient status of patients that participated in the NUTRIENT II trial (NCT02378948). The NUTRIENT II was a multicenter randomized controlled trial that investigated the influence of direct start of oral feeding with the standard of care (jejunostomy tube feeding and 5 days nil-by-mouth) on postoperative recovery following a minimally invasive Ivor–Lewis esophagectomy with intrathoracic anastomosis [22]. The NUTRIENT II database used for the purpose of the current study was already fully constructed and anonymized by the researchers. Therefore, additional informed consent was not obtained.

In the current analysis, only NUTRIENT II patients in which surgery was performed in the Catharina Hospital and had a postoperative evaluation of micronutrients were included. Standard postoperative micronutrient follow-up included evaluation of vitamins B1, B6, B12 and D, whereas a majority of these patients also had an evaluation of folate, methyl malonic acid (MMA) and homocysteine. Evaluation of iron, ferritin, transferrin, hemoglobin (Hb) and mean corpuscular volume (MCV) was carried out on indication. Nutritional supervision was given to every patient during the pre- and postoperative phase by a registered dietitian and focused on changing eating habits and not on vitamin supplementation. Mineral or multivitamin supplementation was not standard of care by the treating physician. 

The study protocol was approved by the institutional review board of the Catharina Hospital, Eindhoven, the Netherlands (nWMO-2019.010).

### 2.3. Definitions

Deficiencies were defined as a value below or above the normal cut-off value as defined by the local clinical laboratory. More specifically, anemia was defined as a Hb level <7.5 mmol/L in women and <8.5 mmol/L in men. In anemic patients, microcytic, normocytic and macrocytic anemia were defined as MCV <80 fL, 80–100 fL and >100 fL, respectively. A true iron deficiency was defined as a serum iron level <10µmol/L and/or ferritin <30 µg/L. A true vitamin B12 deficiency was defined as a serum vitamin B12 level <140 pmol/L and/or a serum MMA >430 nmol/L. The latter is an indicator for assessing functional vitamin B12 levels [23]. Hyperhomocysteinemia, which is associated with low serum vitamin B6, B12 and folate levels [23,24], was defined as a homocysteine serum level >15 µmol/L. Vitamin D (25-Hydroxyvitamin D) deficiency was defined as a serum level <50 nmol/L.

### 2.4. Statistical Analysis

Statistical analyses were performed using the IBM Statistical Package for Social Science (SPSS^®^) software tool (version 25.0). Continuous variables are presented as mean (standard deviation) or median (lower–upper quartile) depending on the normality distribution of data.

## 3. Results

### 3.1. Descriptive Literature Review

Three studies that investigated micronutrient deficiencies in patients following an esophagectomy were identified. The complete study characteristics are summarized in Table 1. In the study (*n* = 187) by van Hagen et al. it was reported that the prevalence of vitamin B12 deficiency in patients at a median of 6.0 months (interquartile range (IQR) 3.5–9.9) to 20.8 months (IQR 14.3–29.7) following an esophagectomy was higher than in the non-operated population [25]. Thirteen of 20 patients with a vitamin B12 deficiency were anemic, while no patients suffered from megaloblastic anemia. Moreover, in the prospectively followed patients that underwent an esophagectomy and were not vitamin B12 deficient at baseline, the estimated one-year incidence was 18.2%.

Heneghan et al. (*n* = 45) reported that 89.7% of patients had at least one fat-soluble vitamin deficiency at 18 to 24 months postoperatively [26]. In these patients, vitamin A deficiency was most common, followed by vitamin E, vitamin D and iron deficiency, respectively (Table 1). The prevalence of vitamin B12 deficiency in these patients was not reported. Moreover, it was reported that serum vitamin A and E levels were significantly decreased at 18 to 24 months after surgery compared to preoperatively, while vitamin B12 and D levels did not change significantly. Despite the fact that this study included esophagectomy (*n* = 30) and gastrectomy (*n* = 15) patients, there were no significant differences between both cohorts with respect to any of the micronutrient values.

The study (*n* = 75) by Elliott et al. reported a substantial number of patients with a vitamin D deficiency both one and two years after surgery. However, it was also reported that serum vitamin D levels did not change significantly [27]. Despite no significant postoperative change in serum vitamin D, calcium or phosphate levels, it was reported that bone mineral density was significantly decreased at one year and two years after surgery, while osteoporosis was detected in 38% and 44% of patients, respectively. In addition, serum vitamin A levels were significantly decreased at both one and two years after surgery, whereas vitamin E levels did not change significantly. However, the prevalence of vitamin A and vitamin E deficiency in these patients was not reported. 

In conclusion, despite the fact that the assessment of micronutrients varied across the limited amount of studies and that the findings were inconsistent, it was shown that micronutrient deficiencies are common in patients following an esophagectomy. 

### 3.2. Characteristics of the Study Group

A total of 95 patients in the NUTRIENT II trial that underwent an esophagectomy with a curative intention between October 2015 and May 2018 at the Catharina Hospital, Eindhoven, the Netherlands were identified. Of these patients, 12 were excluded from the analyses due to a lack of postoperative micronutrient follow-up on account of disease progression prior to the first blood workup (*n* = 11) and an unknown reason in one patient. Thus, in 83 patients, postoperative micronutrient data were available during follow-up. Preoperative micronutrient data were available in only two of 83 patients. Therefore, preoperative values were not analyzed. Patients were predominantly male (79.5%) and mean age was 65 years (standard deviation (SD) 7.7). Median preoperative body mass index (BMI) was 25.3 kg/m^2^ (IQR 23.6–28.7) and median weight loss at 3 months after surgery was 7.1% (IQR 2.5–9.7). Complete patient characteristics are listed in Table 2.

### 3.3. Trends of Micronutrient Values Following an Esophagectomy

A flow-chart depicting the follow-up process is shown in Figure 1 and the corresponding micronutrient values are summarized in Table 3. In addition to the standard micronutrient follow-up as previously described, micronutrients such as magnesium, calcium, phosphate and vitamin A, as well as macronutrients such as albumin and lipids (triglyceride and total cholesterol) were incidentally measured but no deficiencies were detected in these patients. Moreover, except for one patient that had a mildly decreased serum vitamin B1 level at 18.5 months after surgery, no additional deficiencies were detected in vitamins B1 and B6 during follow-up. 

Sixty-five of 83 patients (78.3%) had at least one and 31 of 83 patients (37.3%) had more than one micronutrient deficiency at the first micronutrient follow-up (median 6.1 months after surgery (IQR 5.4–7.46)). Of these, vitamin D deficiency was most common (50.6%), followed by a true—as previously defined—iron deficiency (42.2%), folate (28.8%) and a true vitamin B12 deficiency (18.3%), respectively (Table 3). Furthermore, a substantial number of patients were anemic, while MCV was decreased or increased in only two of the anemic patients (one each for both). 

At the second micronutrient follow-up (median 16.5 months after surgery (IQR 12.2–22.1)), 24 of 59 patients (40.7%) had one or more micronutrient deficiency, while ten of these patients (16.9%) had two or more. More specifically, in eleven patients at least one micronutrient deficiency that was previously detected at the first measurement persisted, while in ten patients a new deficiency developed. In three patients this occurred despite supplementation on indication, however, eight patients did not directly receive supplementation because they were considered as mild deficiencies or because these were not initially recognized. Most common deficiencies at the second measurement included a true iron deficiency (28.6%), folate (20.8%), vitamin D (19.3%) and a true vitamin B12 deficiency (5.1%), respectively (Table 3). Similar to the first micronutrient follow-up, a substantial amount of patients were anemic, while MCV was increased in only one of the anemic patients. 

At the third micronutrient follow-up (median 24.8 months after surgery (IQR 19.4–33.1)), subsequent data were available in 18 patients. Of these, five already had at least one micronutrient deficiency at the second measurement as opposed to six patients in which a new deficiency developed. Consequently, 11 patients (61.1%) had at least one micronutrient deficiency, while seven of these patients (38.9%) had two or more. A true iron deficiency (31.3%) was most common in these patients, followed by folate (23.5%) and vitamin D deficiency (11.8%). Three patients were anemic, while in two of the anemic patients MCV was decreased or increased (one each for both). During the follow-up period, most deficiencies normalized following supplementation.

## 4. Discussion

To date the literature on micronutrient deficiencies following an esophagectomy is ambiguous. However, the current study demonstrated that the majority of our patients had at least one micronutrient deficiency as early as six months following an esophagectomy. This included vitamin B12, vitamin D, folate and iron deficiency. Firstly, the finding of vitamin B12 deficiency was in line with what was reported by van Hagen et al. Proposed mechanisms include (1) reduced dietary intake of vitamin B12 due to anatomic changes leading to early satiety, (2) reduced digestion of vitamin B12 from food due to reduced acid secretion and reduced length of time food is exposed to this acid in the gastric conduit, and (3) reduced production of intrinsic factor [25,28]. On the contrary, while the prevalence of vitamin B12 deficiency was not reported, Heneghan et al. reported no significant change in serum vitamin B12 levels. However, this may due to the fact that an increase in MMA is a more specific and early metabolic marker than serum vitamin B12 levels for detecting effective vitamin B12 deficiency [23,25].

The high prevalence of vitamin D deficiency in this study was in line with what was reported by Heneghan et al. and Elliot et al. However, in both studies it was reported that the preoperative prevalence of vitamin D deficiency was also high and that serum vitamin D levels did not significantly change after surgery. Indeed, vitamin D deficiency is currently a common problem in the general population and it may well be that an esophagectomy does not induce additional vitamin D deficiencies [29]. Nonetheless, similar to other types of gastric surgery [17,21], Elliot et al. reported that even despite the lack of significant postoperative changes in serum vitamin D, calcium or phosphate levels, patients that underwent an esophagectomy were at risk of osteoporosis. 

On the other hand, the finding of folate deficiency in this study was not described in previous studies. Folate deficiency is mainly associated with reduced intake of folate-rich foods, such as dark leafy green vegetables [28]. Moreover, folate and/or vitamin B12 deficiency frequently lead to hyperhomocysteinemia, which was commonly detected in this study and is a risk factor for cardiovascular diseases [23,24].

The high prevalence of iron deficiency in this study was in line with the results from Heneghan et al. and may be explained by (1) inadequate oral intake and (2) the lack of acidity that results in impaired conversion of ingested ferric iron to absorbable ferrous iron in the duodenum and proximal jejunum [19]. In addition, iron and ferritin values may also be affected by acute or chronic inflammation and liver impairment [30]. However, inflammation parameters were routinely measured at the first micronutrient follow-up and were normal in all patients. While liver function was only determined on indication during the postoperative follow-up, we expect that it is unlikely that ferritin and iron were (still) influenced in the absence of inflammation.

Interestingly, while the prevalence of anemia in the current study was high, MCV was normal in the majority of anemic patients, which was similar to previous findings by van Hagen et al. This may be due to the concurrent presence of iron, vitamin B12 and/or folate deficiency in most of these patients and early detection and preventive measures (e.g., supplementation or other nutritional interventions) [23]. Of note, while anemic patients were predominantly male, no significant differences were found in mean values of iron, ferritin, vitamin B12 and MMA between men and women. However, relatively more men had an iron deficiency, but this difference was not significant (40% vs. 16.7%, *p* = 0.129). On the other hand, similar rates were found for ferritin (*p* = 0.972), vitamin B12 (*p* = 0.967) and MMA (*p* = 819). It may also be that preoperatively, the depletion of micronutrient deposits occurred more frequently in men, but preoperative micronutrient parameters were not available in the present study.

Altogether, it seems that similarly to bariatric surgery—in which the anatomy of the upper gastrointestinal tract is deliberately altered to reduce intake and absorption—micronutrient deficiencies may develop in patients following an esophagectomy. Due to the anatomical changes to the upper gastrointestinal tract to replace the resected esophagus, adverse gastrointestinal symptoms, such as nausea, vomiting, regurgitation, diarrhea, dumping or loss of appetite and reduced intake are common. While these symptoms may also contribute to the development of micronutrient deficiencies, exact mechanisms involved in the development of micronutrient deficiencies following an esophagectomy, as well as the decrease in bone mineral density are still not fully understood and may well be dissimilar [13,15,25,27]. However, similarly to gastric surgery, it has been reported that pancreatic insufficiency also contributes to the postoperative morbidity in patients following an esophagectomy [26,27].

Vitamins such as vitamin A, E and K are not routinely determined after an esophagectomy, but are denoted as important vitamins in patients undergoing bariatric surgery. In particular during pregnancy, when daily requirements are higher due to fetal and maternal development, there is an increased risk of developing micronutrient deficiencies and complications [13,16]. Whether or not these vitamins also play an important role in esophageal cancer patients is questionable, considering that esophageal cancer patients are predominantly male and elderly. 

This study has some strengths and limitations. A prospectively maintained database with complete clinical parameters from a randomized controlled trial was used to identify and include patients. However, the data were collected from a single center database and preoperative data on micronutrient status was not available. Therefore, we cannot ascertain whether these deficiencies were already present or developed due to the esophagectomy. Furthermore, despite the fact that it was not standard of care to provide supplementation, rather only when a micronutrient deficiency was detected, occult over the counter supplementation could not be ruled out. This represents an important limitation as it may be that the true incidence of micronutrient deficiencies was underestimated in this cohort. Moreover, at inclusion 55% of patients reportedly consumed alcohol on a daily basis. However, additional data on the quantity and whether patients still consumed alcohol during the preoperative and postoperative phase were missing. Therefore, we cannot determine the effect of alcohol use as a potentially confounding factor on the occurrence of micronutrient deficiencies. 

Nevertheless, based on the findings of the current study and literature to date on micronutrient deficiencies following an esophagectomy, and the more substantial evidence in the long-term follow-up of bariatric surgery patients, it is recommended that patients undergoing an esophagectomy for cancer are routinely screened for micronutrient deficiencies and supplementation provided when needed [13,14,15,16]. Even though most deficiencies normalized following supplementation on indication, micronutrient deficiencies were still common during long-term follow-up in the present study. This was due to the fact that new deficiencies developed or because patients did not yet receive supplementation. Therefore, it is to be considered whether or not oral multivitamin and mineral supplementation should be standardly advised to these patients. 

## 5. Conclusions

In conclusion, micronutrient deficiencies are common in patients as early as 6 up to 24 months following an esophagectomy with gastric conduit reconstruction for cancer. Therefore, it is recommended that these patients are routinely checked and supplementation should be provided when needed to prevent symptoms on the long-term.

## Figures and Tables

**Figure 1 nutrients-12-00778-f001:**
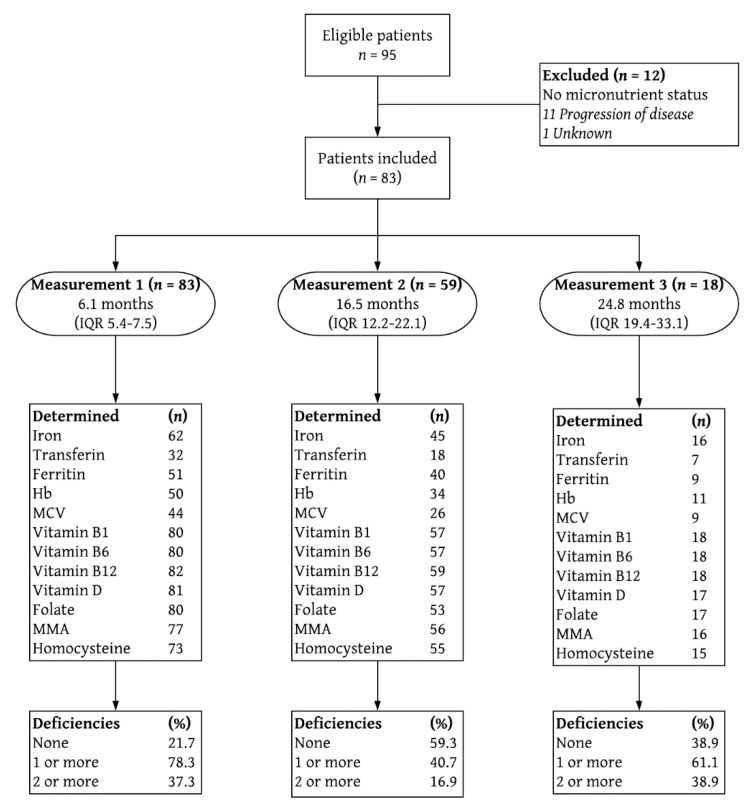
Flowchart depicting study inclusion and follow-up measurements. Deficiencies are % of total. IQR interquartile range; Hb hemoglobin; MCV mean corpuscular volume; MMA methylmalonic acid.

**Table 1 nutrients-12-00778-t001:** Overview of the current literature on micronutrient deficiencies following esophagectomy.

Author	Country	Year	Study Period	Study Design	Study population					Type of Surgery		Follow-up Period		Micronutrient Assessment	Prevalence of Deficiencies after Surgery
					Size	Age		Gender							
					*n*	Years	(SD) or IQR	Male	%	Esophagectomy	Gastrectomy	Months	[IQR]		
van Hagen et al.	Netherlands	2017	August 2010–July 2012	Group A: single center, cross-sectional cohort study	99	62	38–79	73	74	99	–	19.4	[13.5–30.1]	Vitamin B12	11%
				Group B: double center, prospective cohort study	88	63	19–79	63	72	88	–	6.4	[1.6–10.8]	Vitamin B12	10.2%
Heneghan et al.	Ireland	2015	January 2013–July 2013	Single center, prospective cohort study	45	63.3	(8.9)	30	68	30	15	23.1	[18,19,20,21,22,23,24,25,26,27,28]	Vitamin A Vitamin B12 Vitamin D Vitamin E Iron	81.5% missing 44.4% 61.5% 46–49%
Elliot et al.	Ireland	2019	2000–2014	Single center, retrospective cohort study	75	60.6	(9.6)	58	77	75	–	43.4	Missing	Vitamin A Vitamin D Vitamin E	missing 53%–70% ^†^ missing

**Legend:** values mean (standard deviation) or median [lower-upper quartile]. † Serum vitamin D value <50 nmol/L.

**Table 2 nutrients-12-00778-t002:** Baseline characteristics of study population.

Clinical Characteristics	(*n* = 83)	
Sex (male)	66	(79.5)
Age at inclusion (SD)	65	(7.7)
ASA score		
I	7	(8.4)
II	63	(75.9)
III	13	(15.7)
Preoperative BMI	25.3	(23.6–28.7)
Alcohol use (at inclusion)		
No	18	(21.7)
Daily	46	(55.4)
Weekly	5	(6.0)
Monthly	14	(16.9)
Any comorbidity	54	(65.1)
pTNM stage		
Stage 0	29	(34.9)
Stage I	20	(24.1)
Stage II	19	(22.9)
Stage III	15	(18.1)
Histology type		
Adenocarcinoma	68	(81.9)
Squamous-cell carcinoma	15	(18.1)
Tumor localization		
Mid esophagus	4	(4.8)
Distal esophagus	51	(61.4)
Esophagogastric junction	28	(33.7)
Neo-adjuvant treatment	76	(91.6)
NUTRIENT II treatment allocation		
Direct oral feeding	42	(50.6)
Standard of care	41	(49.4)
Any 30-day postoperative complication	63	(75.9)
BMI at discharge	25.0	(23.5–28.7)
BMI 3 months postoperatively	23.3	(22.0–26.9)

**Legend:** values are absolute value (percentage), mean (standard deviation) or median (lower–upper quartile). BMI body mass index.

**Table 3 nutrients-12-00778-t003:** Hematological and nutritional profile in postoperative follow-up.

		Measurement 1						Measurement 2						Measurement 3					
		6.1 Months (5.4–7.5)						16.5 Months (12.2–22.1)						24.8 Months (19.4–33.1)					
Micronutrient	Normal Value	Total			Abnormal Value			Total			Abnormal Value			Total			Abnormal Value		
Standard		(*n*)	Value		(%)	Value		(*n*)	Value		(%)	Value		(*n*)	Value		(%)	Value	
**Vitamin B1**	70–200 nmol/L	80	132	(31)	–	–	–	57	137	(37)	–	–	–	18	136	(31)	5.6	68	–
**Vitamin B6**	35–110 nmol/L	80	75	(58–95)	–	–	–	57	81	(64–110)	–	–	–	18	67	(54–90)	–	–	–
**Vitamin B12**	140–700 pmol/L	82	240	(200–323)	6.1	110	(89–130)	59	310	(220–470)	1.7	130	–	18	240	(210–410)	–	–	–
**Vitamin D**	>50 nmol/L	81	48	(36–68)	50.6	36	(29–42)	57	63	(55–74)	19.3	34	(33–41)	17	63	(55–77)	11.8	47	(45–49)
**Folate**	>10 nmol/L	80	14	(10–22)	28.8	7	(6–9)	53	20	(13–35)	20.8	8	(7–9)	17	14	(10–24)	23.5	7	(5–8)
**MMA (↑)**	0–430 nmol/L	77	230	(186–302)	14.3	569	(482–692)	56	195	(168–246)	5.4	475	(470–611)	16	220	(173–403)	12.5	721	(667–775)
**Homocysteine (↑)**	<15 μmol/L	73	13	(11–16)	35.6	18	(16–20)	55	12	(10–14)	21.8	18	(16–20)	15	13	(10–20)	33.3	21	(18–24)
**On indication**																			
**Hb**	(♂) 8.5–11.0 mmol/L	50	8.6	(8.0–8.9)	32	7.7	(7.2–8.3)	34	8.6	(8.1–8.9)	26.5	7.9	(7.7–8.3)	11	8.7	(7.7–9.3)	18.2	7.6	(7.2–7.9)
	(♀) 7.5–10.0 mmol/L		8.4	(8.0–8.6)	–	–	–		8.7	(7.9–9.1)	–	–	–		8.4	(7.5–8.7)	9.1	6.7	–
**MCV (↑)**	80–100 fL	44	91	(87–93)	4.5	80	(79–80)	26	93	(89–96)	11.5	103	(102–108)	9	90	(88–91)	11.1	68	–
**MCV (↓)**					4.5	101	(100–101)				–	–	–				11.1	101	–
**Iron**	14–35 μmol/L	59	14	(12–17)	35.5	10	(8–13)	45	16	(14–20)	17.8	9	(7–12)	16	16	(12–19)	25	10	(5–12)
**Transferrin**	2.2–3.6 g/L	32	2.6	(2.2–2.9)	18.8	2.1	(2.0–2.2)	18	2.5	(2.1–2.8)	33.3	2.0	(1.6–2.1)	7	2.8	(2.0–3.3)	28.6	2.0	(1.9–2.0)
**Ferritin**	30–400 μg/L	51	76	(32–180)	19.6	16	(14–23)	40	94	(33–169)	20	14	(11–24)	9	32	(25–100)	22.2	19	(13–25)

**Legend:** values are mean (standard deviation) or median (lower–upper quartile). Abnormal values are % of determined (*n*) and always indicate a decreased value unless otherwise indicated (↓ decreased; ↑ increased). Hb hemoglobin; MCV mean corpuscular volume; MMA methylmalonic acid.

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
