# Peer review of "Micronutrient Deficiencies Following Minimally Invasive Esophagectomy for Cancer"

_nutrients, 2020, doi:10.3390/nu12030778_

Round 1

Reviewer 1 Report

Review micronutrients deficiencies

This article gives data on longterm follow-up of the micronutrient status after esophagectomy. Especially the finding that so long after an esophagectomy still deficiencies are found is a very important finding. I like to underline that in the key words or mention it separate as a highlight.

No data are given on the fat soluble vitamin K. In your patients that might not be a problem, but during pregnancies it can be a problem. Maybe you can say a short word about that problem in pregnancy. See attached papers (1, 2).

In table 2 in the upper line starting with clinical characteristics the number is written above the % line. You might adapt that.

In the important table 3 you have mentioned MMA and Homocysteine levels as Deficiency, maybe you can better call that whole row “Abnormal” ( deficient/too high )

In the conclusion I should not only tell that the problems are there after 6 months but you might underline your finding that you still have problems even after 24 months. That is important because on the moment after a Roux-Y procedure the rule is that the woman might become pregnant after two years. However the deficits are still there and can give problems to the baby f.i an intracranial bleeding in the unborn baby due to vitamin K deficiency. It underlines your important finding that the deficiencies can stay for years.

  1. Van Mieghem T, Van Schoubroeck D, Depiere M, Debeer A, Hanssens M. Fetal cerebral hemorrhage caused by vitamin K deficiency after complicated bariatric surgery. Obstetrics Gynecology. 2008;112:434-6.
  2. Jans G, Guelinckx I, Voets W, Galjaard S, van Haard P, vansant G, et al. Vitamin K1 monitoring in pregnancies after bariatric surgery: a prospective cohort study. Surg Obes Relat Dis 2014;10(5):885-90.

Author Response

Dear reviewer,

First of all we would like to thank you for the interest in and critical appraisal of our manuscript. Please find our response to your comments below.

Ad 1. Thank you for this valuable comment regarding the finding of deficiencies in long-term follow-up after an esophagectomy. While we agree with the you that reporting long-term consequences is important, we had chosen to not highlight this data since we only have data in 18 patients at 24 months after esophagectomy. We have now also addressed the current long-term results more extensively in the discussion (page 10, line 253-256).

Ad 2. You comment on the fact that data on vitamin K was lacking in our cohort and this might be a problem in pregnancies but not in the current cohort. Since postoperative evaluation of vitamin K is not standard of care in our hospital, we cannot report on vitamin K deficiency. However, we have added this in the discussion and elaborated on vitamins A and E as well (page 10, line 230-235).

Ad 3. Thank you for poinintg out a typographical error in Table 2. This has been corrected (page 5).

Ad 4. Thank you for your suggestion to mention the aberrant MMA and Homocysteine levels in Table 3 as abnormal rather than as a deficiency since these values are increased. We agree that this is more fitting even though these are also indicative of deficiencies. We have changed this and clarified this in Table 3 within the corresponding legend (page 8).

Ad 5. Similar to your first comment that long-term presence of deficiencies should be mentioned in the conclusion, we have added this in the abstract (page 1, line 21-23) and conclusion (page 10, line 259-260).

Reviewer 2 Report

The manuscript is well written and informative. One suggestion to the authors for improvement is to elaborate in the introduction and discussion on the primary reasons why esophagectomy is associated with micronutrient deficiencies; chief among which is the use other sections of the GI tract, stomach or sometimes large intestine, to reconstruct the conduit which alters normal nutrient digestion and absorption.

Author Response

Dear reviewer,

First of all we would like to thank you for the interest in and critical appraisal of our manuscript. Please find our response to you comment below.

Ad 1. You suggest to elaborate on the primary reasons why an esophagectomy is associated with micronutrient deficiencies. Thank you for this valuable suggestion. We have provided additional information in the introduction (page 1, lines 39-45) and discussion (page 9, line 221-229).

Reviewer 3 Report

Interesting study. There is a lack of literature in this field, so this article will help support practice changes in this area.

Overall good, but needs some more detail. 

Your systematic review needs to be fully documented  - please include how many articles your literature review identified and the exclusion process utilised to demonstrate how you ended up with only 3 studies. 

A few points I think need clarification:

You describe alcohol use in your demographics - what was the quantity - ?did this impact on nutritional status? With 70% consuming alcohol daily this may be a consideration - do those drinking to excess have a higher incidence of micro-nutrient deficiencies - can you subgroup the data? If unknown needs addressing in the limitations. 

You need to discuss the limitations of biochemical monitoring within this study. For instance Ferritin and Iron are influenced by acute phase / liver impairment / inflammation and elevated ESR levels are seen in malignancy. Were any other blood results are available to identify any confounders - liver function tests, CRP, ESR, White cell count etc?? 

Why do you think there is a higher incidence of anaemia in men? Is this significant?

Your rate of Vitamin D deficiency reduces with time? This is highest in the first measurement, when patients will have just completed neo-adjuvant chemotherapy followed by surgery. One could hypothesise that their sunlight exposure may be reduced during the months they are going through this. Or do we assume they were supplemented in response to the the first deficiency result, you have stated in the discussion (line 211) that supplementation would have been provided in the episode of deficiency?

You mention preventative measures in line 198 - please expand?

Is supplementation sufficient - do you think bone density assessment is warranted?

Have you considered secondary pancreatic exocrine insufficiency as a contributory factor? This has been raised in Elliot et al, and an incidence study published (Huddy et al, 2013. Dis Esophagus 26:594–597

Not knowing whether patients are on supplements or not is a serious flaw in the data, and should be acknowledged more comprehensively. However, if there are a number of patients on oral supplements, this means your study has likely under-estimated the incidence of deficiency.  

Author Response

Dear reviewer,

First of all we would like to thank you for the interest in and critical appraisal of our manuscript. Please find our response to your comments below.

Ad 1. You justly point out that full documentation of the systematic review is warranted. However, we performed a descriptive literature review. To avoid confusion we have removed the word “systematic” in the manuscript.

Ad 2. You comment on alcohol use and quantity in our cohort. We noticed the percentage of daily alcohol consumption to be based on the cohort of patients consuming alcohol, rather than the complete cohort. This has been altered in Table 2 (page 5). While alcohol consumption was only determined at baseline, additional data on the quantity and whether or not patients still consumed alcohol during the preoperative and postoperative phase were missing. Therefore, we cannot determine the effect of alcohol use as a potentially confounding factor on the occurrence of micronutrient deficiencies. Since 55% of patients reportedly consumed alcohol daily but the quantity of alcohol consumption is lacking, this has been addressed as a limitation (page 10, line 244-248).

Ad 3. You comment on the limitations of biochemical monitoring within this study. Thank you for this is interesting point. Indeed, postoperative measurement of liver function and inflammation parameters, including CRP and white cell count, is routinely performed. While these parameters are often elevated directly postoperatively, discharge criteria denote that patients are only discharged when these parameters are declining or have normalized. During the postoperative follow-up period the liver function is only determined on indication. Therefore we cannot ascertain whether or not this may have influenced the iron or ferritin values. However, inflammation parameters were routinely measured at the first micronutrient follow-up and were not elevated in all patients. On the other hand, while iron may be depleted, it is to be expected that ferritin levels are elevated and not decreased in case of (chronic) inflammation (Lopez et al, 2016. Lancet 387(10021):907–916). Hence, we believe that it is unlikely that Ferritin and Iron were still influenced by inflammation or liver impairment during follow-up. We have added more discussion on this (page 9, line 203-207).

Ad 4. You comment on the higher incidence of anemia in men in the current cohort. Indeed, a higher incidence in anemia rate is seen in men, but there was no significant difference in mean values of vitamin B12, MMA, iron and ferritin between men and women. While we did not identify any significant factors contributing to this finding, relatively more men had an iron deficiency (40% vs. 16.7%, p=0.129). On the other hand, similar rates were found for ferritin (p=0.972), vitamin B12 (p=0.967) and MMA (p=819). It may also be that in this cohort the depletion of preoperative micronutrient deposits occurred more frequently in men. However, no preoperative micronutrient parameters were available. More discussion on this has been added to page 9, line 212-218.

Ad 5. You comment on the declining rate of Vitamin D deficiency from measurement 1 to measurement 3. Potentially the sunlight exposure of patients may be reduced during these months due to treatment or seasonal factors. However, as no data on preoperative measurement of micronutrient values (including vitamin D) were available, we cannot describe the complete trend. Since all patients with a deficiency are routinely supplemented as mentioned in the results (page 6, line 167-168) and discussion (page 10, line 253-254), most deficiencies (including vitamin D) normalized. Therefore, the declining rate of vitamin D deficiency is most likely due to the supplementation on indication. Hence, we have not added more discussion on this.

Ad 6. You ask to elaborate on preventive measures mentioned in the discussion. With preventive measures we mean nutritional supervision (e.g. supplementation on indication). This applies in particular to vitamin B12 deficiency because methylmalonic acid (MMA) is an early detection marker. Hence, providing supplementation to patients with elevated MMA levels may decrease incidence of complications due to vitamin B12 deficiency. We have clarified this on page 9, line 209-212.

Ad 7. You ask if supplementation is sufficient or bone mineral density (BMD) assessment is warranted. In our cohort BMD was not assessed, therefore we cannot ascertain whether BMD declined in these patients. However, while assessment of BMD is useful to identify patients that are more at risk for osteoporosis and fractures, therapeutic consequences are the same. Among other things, this adds to the question whether or not it is to be considered that oral multivitamin and mineral supplementation should be standardly advised to patients following an esophagectomy. However, as mentioned by Elliot et al, data on the extent to which bone remodeling after esophagectomy is related to postoperative factors requires further study. Therefore, we believe more research is needed before we can provide a definitive answer on whether or not BMD assessment is warranted. Due to the lack of data we have not altered the current discussion on this.

Ad 8. You ask whether we considered secondary pancreatic exocrine insufficiency as a contributory factor to development of micronutrient deficiencies. Thank you for pointing this out to us. Exact mechanisms involved in the development of micronutrients are still not fully understood. However, we agree with you that further elaboration is warranted. We have added more discussion on page 10, line 227-229.

Ad 9. You comment that not knowing whether patients use supplements is a serious flaw in the data and should be acknowledged more comprehensively. We agree with you and have elaborated more on this limitation as a potential confounding factor (page 10, line 242-244).

Round 2

Reviewer 3 Report

All concerns adequately addressed in text.